# The influence of hip revision stem spline design on the torsional stability in the presence of major proximal bone defects

Julius M. Boettcher[1]*, Kay Sellenschloh[1], Gerd Huber[1], Benjamin Ondruschka[2], Michael M. Morlock[1]

1 Institute of Biomechanics, Hamburg University of Technology, Hamburg, Germany, 2 Institute of Legal Medicine, University Medical Center Hamburg-Eppendorf, Hamburg, Germany

* julius.boettcher@tuhh.de

## Abstract

### Background

Despite the success of primary total hip arthroplasty, the number of revisions remains high. Infection, aseptic loosening, periprosthetic fractures and dislocations are the leading causes of hip revision. Current revision stem designs feature a tapered body with circumferential placed longitudinal thin metal splines that cut into the femoral cortex of the diaphysis to provide axial and rotational stability. Modifications to the spline design may help improve primary stability in various bone qualities. The purpose of this study was to evaluate whether the rotational stability of a revision hip stem can be improved by an additional set of less prominent, wider splines in addition to the existing set of splines. It is hypothesized that the additional splines will result in greater cortical contact, thereby improving torsional strength.

### Methods and findings

The ultimate torsional strength of an established modular revision stem (Reclaim®, DePuy Synthes) was compared to a Prototype stem design with two sets of splines, differing in prominence by 0.25 mm. Five pairs of fresh-frozen human femurs (n = 10) were harvested and an extended trochanteric osteotomy was performed to obtain common bone defects in revision. Stems were implanted using successive droptower impacts to omit variability caused by mallet blows. The applied energy was increased from 2 J in 1 J increments until the planned implantation depth was reached or seating was less than 0.5 mm at 5 J impact. The ultimate torsional strength of the bone-to-implant interface was determined immediately after implantation. Image superposition was used to analyze and quantify the contact situation between bone and implant within the femoral canal. Cortical contact was larger for the Prototype design with the additional set of splines compared to the Reclaim stem (p = 0.046), associated with a higher torsional stability (35.2 ± 6.0 Nm vs. 28.2 ± 3.5 Nm, p = 0.039).

**Data Availability Statement:** All relevant data are within the paper and its Supporting information files.

**Funding:** This study was funded by DePuy Synthes. The funders had no role in study design, data collection and analysis, decision to publish, or preparation of the manuscript.

**Competing interests:** I have read the journal's policy and the authors of this manuscript have the following competing interests: MMM is a paid consultant of DePuy Synthes and obtains research support as a Principal Investigator from Ceramtec, DePuy, and Beiersdorf. He obtains speaker's fees from Aesculap, Ceramtec, DePuy, Zimmer, Peter Brehm, Corin, and Mathys and is in the editorial board "Trauma und Berufskrankheit." GH is an associated member of the board of the German Society of Biomechanics.

## Conclusions

A second set of splines with reduced prominence could be shown to improve primary stability of a revision stem in the femoral diaphysis in the presence of significant proximal bone loss. The beneficial effect of varying spline size and number has the potential to further improve the longevity of revision hip stems.

## Introduction

The german arthroplasty registry EPRD documents 177,826 primary THR and 18,145 revision total hip replacements (THR) for 2022 [1], while the National Joint Registry (NJR) for England, Wales, Northern Ireland, the Isle of Man and Guernsey reports, 135.000 revision THRs for the time period between 2003 and 2021 [2]. These high numbers of revisions pose a challenge for surgeons due to reduced bone stock and increasing patient age with declining bone quality [3–6]. With each successive revision, the risk of subsequent revision increases [7].

Aseptic loosening is one of the major causes for hip re-revisions [2] which historically occurred in the second or third decade following primary total hip arthroplasty due to a biological process that weakens the bone-implant interface, mainly due to the immunological effect of wear debris (particle disease). Early aseptic loosening is a preventable complication that occurs when adequate resistance to subsidence or relative torsional motion is not achieved postoperatively [8, 9]. The torsional loads which the implanted revision stem must withstand during a gait cycle are about 26 Nm [10, 11]. Tapered stems with longitudinal splines have shown to resist these loads and show good long-term clinical results, which has led to the development of several design variants [12, 13]. The high revision rates of revision stems, however, indicate that the initial stability still needs further improvement.

The suggested length over which the implant should engage with the bone ranges from as little as 20 mm for tapered fluted designs, to as much as 80 mm for cylindrical uncemented femoral stems [14, 15]. A circumferential press fit achieving indentation of the spines into the cortex, or at least cortical contact of the spines over a certain length, appears to be superior to a three-point scratch fit, but is often difficult to achieve intraoperatively due to the natural curvature of the femur [16]. The hypothesis was that the second set of splines, which engage with the cortical bone during implantation, could significantly increase the contact surface area and therefore provide more resistance to applied torsional load.

The purpose of this study was to evaluate the effect of adding a second set of less prominent splines on the immediate postoperative torsional resistance of revision hip stems and to assess the potential of using the implant-bone contact length as a predictor for direct postoperative stability.

## Material and methods

Five human femur pairs (4 male, 1 female, 69.2 ± 11.0 years) were excised postmortem by the Institute of Forensic Medicine Hamburg and stored below −20°C [17]. This study was approved by the Ethics Committee of the Hamburg Medical Association (2023-300350-WF). Individual patient data could no longer be associated with the specimens used.

CT scans (Incisive CT 128; Philips, Amsterdam, The Netherlands; voxel size: 0.4 × 0.4 × 0.4 mm$^3$) of the femurs together with a calibration phantom (QSA; QRM, Moehrendorf, Germany) were obtained. Hounsfield units were converted to bone mineral density (BMD) in terms of hydroxyapatite concentrations (Structural Insight 3; University Medical Center

Schleswig-Holstein [18], Kiel, Germany). Cortical BMD was determined in a 10 mm-wide ring of the femoral shaft at a distance of 5.8% body height distal to the trochanter minor [19]. An image processing algorithm was used to automatically detect the minor trochanter (threshold: [400–2000 mgHA/cm$^3$] [20]; Matlab 2022a; The Mathworks, Natick, MA). Trabecular BMD was determined based on a spherical region of interest (ROI) in the center of the femoral head (volume: 1000 mm$^3$), which was also automatically detected.

## Proximal bone defects

An extended trochanteric osteotomy (ETO) was performed on the specimens to represent a common residual bone stock situation during revision surgery, preventing proximal bone contact of the revision stem to the femur. The ETO was performed in four steps. First, the femoral head was removed with an L-shaped saw cut to allow access to the femoral canal during subsequent canal reaming (Fig 1A). The linea aspera served as the posterior landmark for the start of the ETO in the distal direction (Fig 1B) [21]. The cut was made at a length of 7.5% of the patient's body height, which corresponds to approximately 12 to 13 cm. The osteotomy was then continued from posterior to anterior over 120°, ending towards the proximal end of the bone (Fig 1C). Cerclage wires were placed distal of the ETO to prevent fractures during subsequent reaming and implantation. The specimens were reamed according to the manufacturer's instructions until the planned prosthesis size was reached. After reaming, the specimens were embedded into a holding pot for later fixation to the testing machine (Technovit 4004; Kulzer GmbH, Wehrheim, Germany). Alignment during embedding of the specimens was ensured by the long axis of the last reamer, which remained in the femoral canal after reaming until the embedding was finished.

## In-vitro experiment

Distal stems of a modular revision system (Reclaim Revision Hip System; Depuy Synthes, Warsaw, IN; stem-length: 140 mm) were implanted in one femur of each pair (n = 5). A

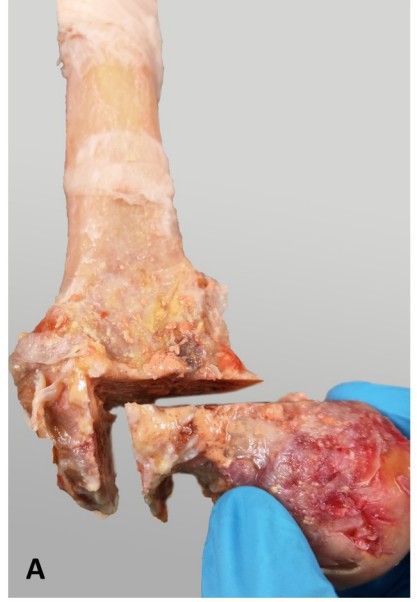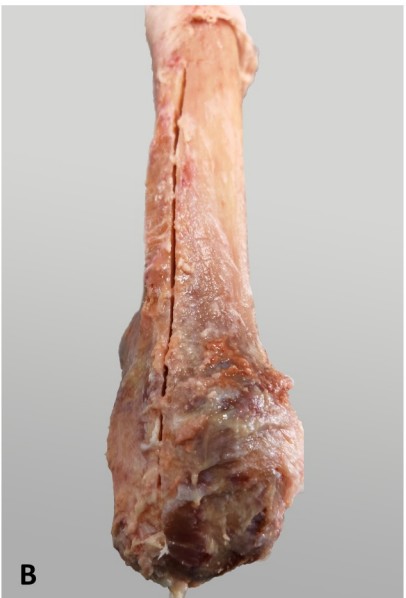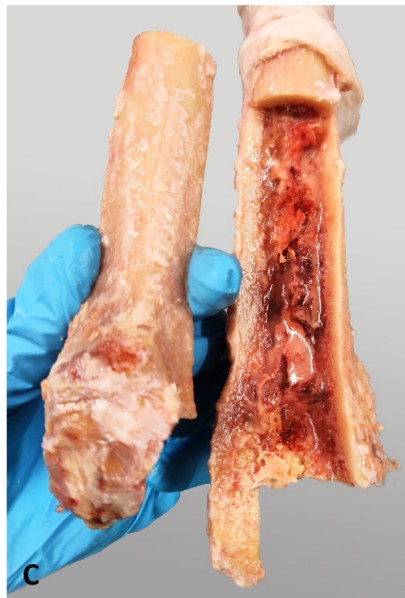

**Fig 1. Simulation of proximal bone defects by performing an ETO.** (A) Removal of the femoral head. (B) First ETO saw cut in distal direction following the linea aspera. (C) Completion of the ETO in proximal direction and removal of the 120° window including the majority of the greater trochanter.

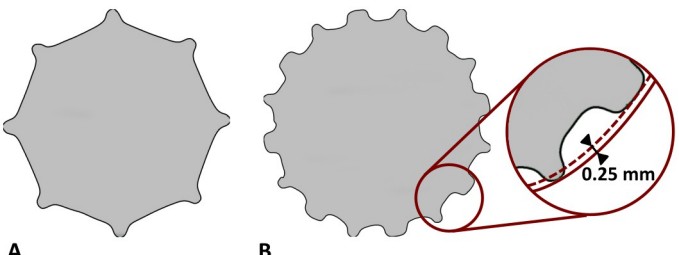

**Fig 2.** (A) Cross-sectional view of the Reclaim implant. (B) Cross-sectional view of the Prototype design. The flat splines are 0.25 mm less prominent than the second set of splines.

Prototype monobloc prosthesis (Depuy Synthes, Warsaw, IN; stem-length: 235 mm) was implanted in the corresponding contralateral femur (n = 5). Powered reaming to the depth determined during planning was performed on both implants according to the respective surgical technique using respective straight helical reamers corresponding to each of the stem designs. The final reaming position was determined by the markings on the instruments. The modular Reclaim stem has eight splines in cross-section (Fig 2A), whereas the Prototype monobloc stem has two different sets of eight longitudinal splines each (one sharp, one flat), with the flat splines being 0.25 mm less prominent (Fig 2B).

Stem implantation was performed using a droptower (Fig 3) for both stem designs by an engineer trained by an experienced surgeon to eliminate variability caused by mallet blows. The bones were aligned with the linear guide of the droptower to ensure axial impaction of the implants. Prototype and Reclaim stems were connected with an adapter and implanted to the final implant position as defined by the reaming process. Impaction forces were recorded with a load cell, placed between the stem adapter and impactor (9333a; Kistler, Winterthur, Switzerland). Forces were recorded at 800 kHz for 50 ms for each impact of the droptower weight.

The position of the implant in relation to the distal femoral ETO cut was recorded during the implantation process using digital image correlation (DIC) (25 fps, FOV: 2752 x 2200 px, marker size: 0.4 mm; ARAMIS 3D Camera; GOM, Braunschweig, Germany).

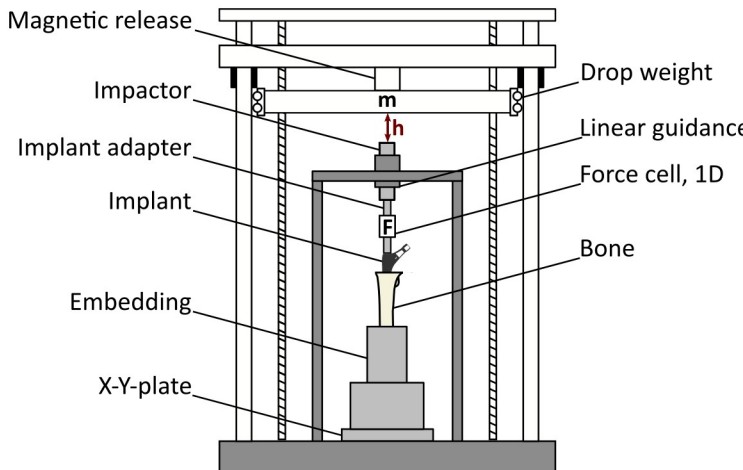

**Fig 3. Schematic drawing of the droptower used for stem impaction.** Dynamic impaction force was recorded with a load cell between stem adapter and impactor.

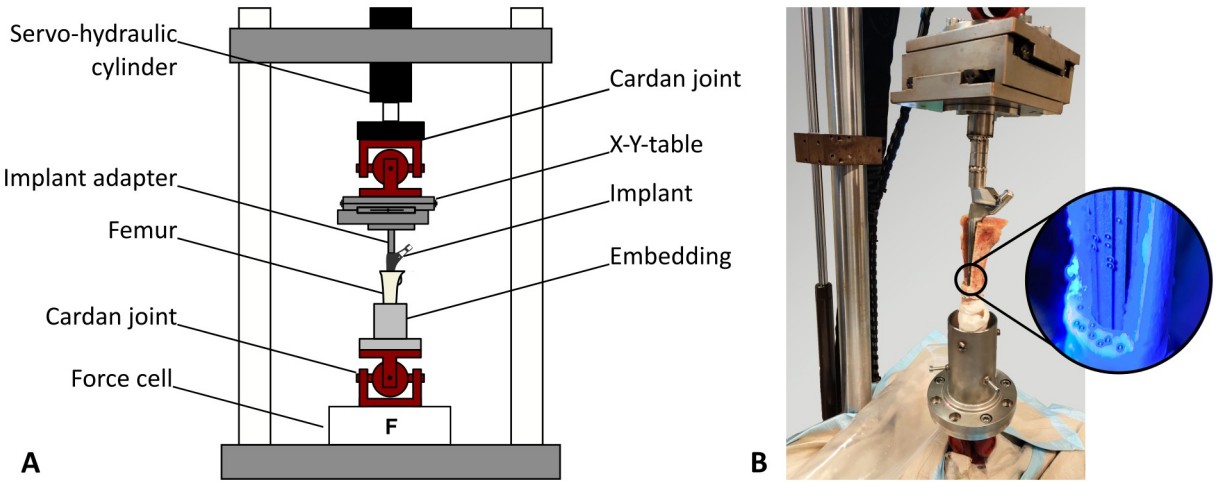

**Fig 4.** (A) Schematic drawing of the test setup. The top-mounted X-Y table allowed the bone-to-implant junction to be mounted in alignment with the axis of the two cardan joints. This allowed axial load and torque to be applied around the implant axis without detrimental limitations. (B) DIC markers were placed on the bone surface and the stem adapter to determine the relative torsion angle between bone and implant.

The initial impaction energy was set to 2 J (drop weight: 5 kg, drop height: 40 mm). If the actual implantation depth, determined from the markings on the stem adapter, did not reach the final position according to the pre-op planning and the incremental seating of one impact was less than 0.5 mm, the drop height was increased to increase the energy level by 1 J. This process was repeated until a maximum energy level of 5 J was reached. Higher energies were not applied to prevent fractures due to excessive forces compared to the forces produced by a "normal" hit with a surgical mallet [22]. This resulted in two criteria for stopping the implantation: either the intended implantation depth according to the preoperative planning was reached, or the incremental seating was less than 0.5 mm per hit at the final energy level of 5 J. Cumulative energy and forces per implantation were then calculated.

Dynamic stem seating of each hit was defined as the change in seating depth (x) divided by the maximum impaction force (F). Exponential functions were used to determine the seating coefficient τ (Eq (1)).

$$f(x) = e^{-\frac{x/F}{\tau}} \tag{1}$$

After implantation, the specimens were mounted in a servo-hydraulic testing machine (MiniBionix II; MTS, Eden Prairie, MN) by two cardan joints to allow a bending-moment free application of the static preload and subsequent torque to failure (Fig 4A). A X-Y table was placed below the upper cardan joint to enable alignment during mounting. This lateral degree of freedom was restricted during the subsequent torque-to-failure test. An axial static preload of 500 N was applied and torque was increased displacement controlled at a rate of 0.5°/s. Peak torsional loads were recorded using the internal force cell of the servo-hydraulic testing machine, and the relative torsion angle was measured based on markers attached to the bone surface and the stem adapter using DIC (same settings as during stem implantation; Fig 4B).

## Contact analysis

The contact situation of the implant in the femoral canal was determined from superimposed volumetric models derived from CT images and 3D laser scans. CT scans of the bones were

obtained initially and after cavity reaming. The cortices of the bones were segmented from the CT images after cavity reaming (threshold: [400–2000 mgHA/cm3], Avizo Lite 2020.2; Thermo Fisher Scientific, Waltham, MA). 3D laser scans of the modular stem and the monobloc implant, as well as scans of the implanted situation in the femur were obtained using a handheld 3D laser scanner (HandySCAN 700; Creaform, Quebec, Canada) with a scan resolution of 0.2 mm.

The polygon models of the bone-implant construct served as a reference to which the implant model and the reamed cavity were aligned (Fig 5; PolyWorks Metrology Suite 2020; InnovMetric Software Inc., Quebec, Canada). Landmarks such as the lesser trochanter, points on the cutting surface of the ETO and unique features on the stem adapter were used for alignment by means of an Iterative Closest Point (ICP) algorithm. The stopping criterion for the ICP algorithm was defined as a residual Root Mean Square Error (RMSE) of less than 0.2 mm, corresponding to the resolution of the 3D laser scanner. Intersections between the implant mesh and the reamed cortical cavity were determined and the corresponding mesh points

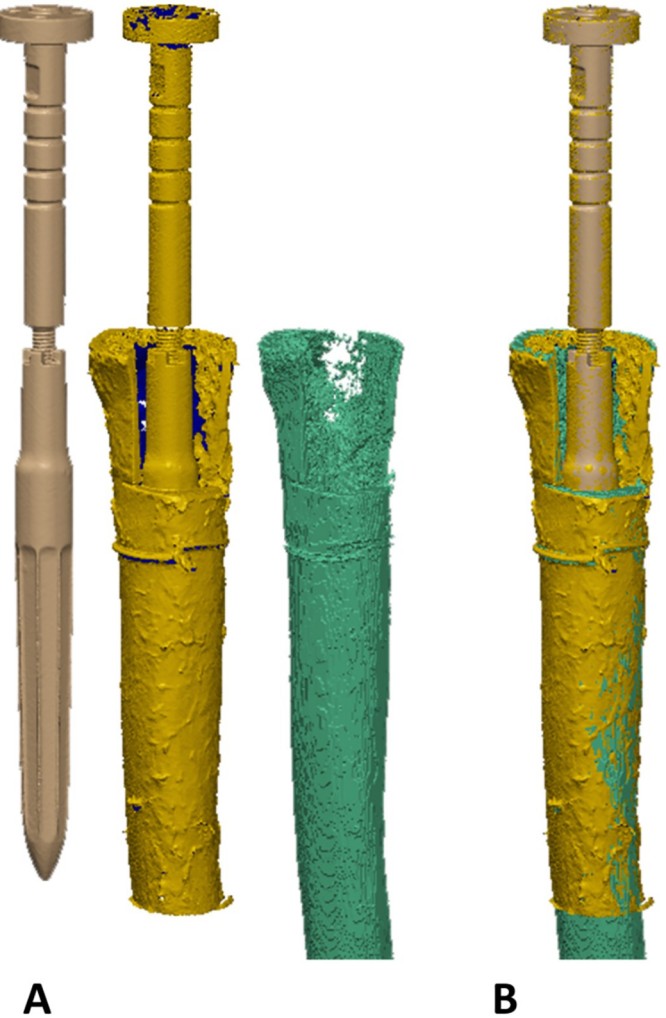

**A**          **B**

**Fig 5. Alignment of the laser scan of a reclaim stem and the segmented cortical bone from the CT scan.** (A) Isolated meshes of the implant, the implant in the implanted state and the reamed cortical bone. (B) Aligned meshes based on the laser scan of the implant in the implanted state. The stopping criterion for ICP alignment was an RMSE of less than 0.2 mm.

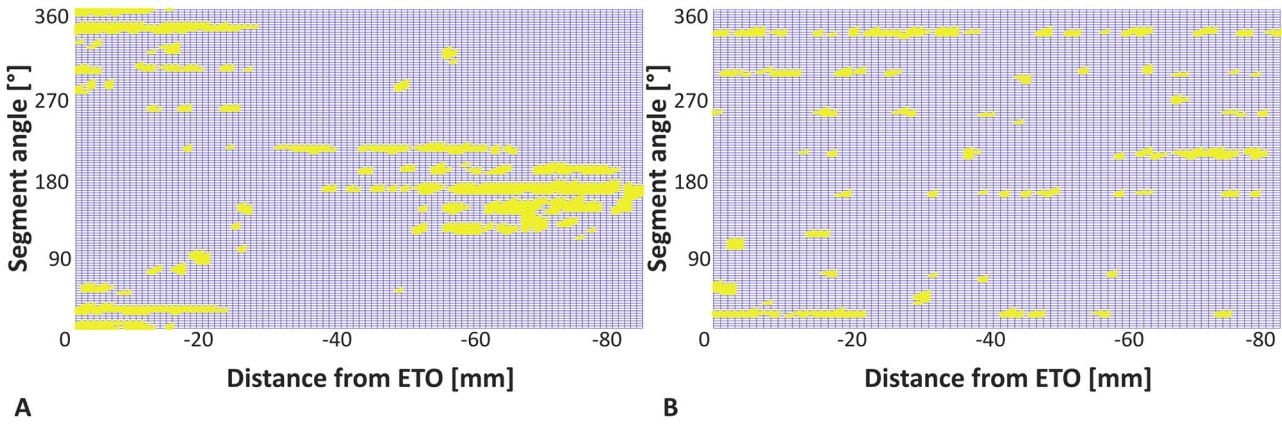

**Fig 6. Representative examples of the contact matrix revealing the intersections (yellow) between the mesh of the implant and the mesh of the reamed cortical bone.** (A) Contact matrix for a Prototype stem. (B) Contact matrix for a stem of the Reclaim design in the contralateral femur. Contact was evaluated between the distal incision of the ETO and the tip of the stem.

were exported as point clouds. The aligned meshes of the segmented bones, all respective contact points defined by overclosure and the laser scans of the corresponding implant were imported as point clouds for further numerical analysis.

The deviation between the achieved final implant position and the intended position according to the preoperative planning and the angular difference between the implant axis and the axis of the reamed cavity before implantation were used to quantify the reaming accuracy. Indentation depth of the splines was evaluated as the overclosure in the x- and y-directions, while the z-axis was defined to coincide with the implant axis. The region of interest (ROI) around the stem surface was divided into 120 sectors of 3˚ opening angle each, starting with 0˚ medially, to determine the location and amount of contact area between the implant and cortical bone. All sectors were searched for contact between implant and cortex in 1 mm increments from the stem tip to the distal incision of the ETO. This resulted in a contact matrix, which was evaluated for coherent contact areas (Fig 6). Specifically, the contact area was evaluated in terms of matrix segments indicating cortical contact for comparison with contact along one sector as a measure of the one-dimensional contact length. In addition, the contact locations were assessed in the anterior-posterior, medial-lateral and proximal-distal direction. Clinically contact length is typically measured intra- and postoperatively from conventional X-rays. The influence of the reaming process on the contact situation and the resulting torsional stability was analyzed using the removed cortical bone volume, calculated as the difference between the femoral cavities before and after reaming (threshold: [-250-400 mgHA/cm$^3$]).

## Statistical analysis

Statistical analysis was performed with a type I error level of 0.05 (SPSS, Version 26; IBM SPSS Statistics, Armonk, NY). Normal distribution and homoscedasticity were tested using Shapiro-Wilk and Levene test statistics. Parametric test statistics were performed to compare the means and test for potential correlations.

## Results

Cortical (1056 ± 62 mgHA/cm$^3$) as well as trabecular BMD (272 ± 49 mgHA/cm$^3$) were similar for both stem design groups ($p_{cortical}$ = 0.893, $p_{trabecular}$ = 0.762, independent t-test).

About 70% of the bones were Dorr type C according to the adjusted definition of Konow et al. [19].

## Implantation process

Required cumulative implantation energy (p = 0.702, dependent t-test) and cumulative implantation force (p = 0.576, dependent t-test) did not show significant differences between the two implant designs. The maximum implantation force was higher for the Prototype design but not significantly (5.042 ± 0.893 kN vs. 4.191 ± 0.422 kN; p = 0.120, dependent t-test). Stem seating characteristics did not show differences between the twoimplant designs (Fig 7A). The mean seating coefficient was higher for the Prototype design (2.09 ± 1.19 vs. 1.64 ± 0.93; not significant p = 0.305, dependent t-test, Fig 7B). A higher seating coefficient is indicative for slower seating.

## Torsional stability

Torsional stability was higher for the Prototype design compared to the Reclaim stem (35.2 ± 6.0 Nm vs. 28.2 ± 3.5 Nm; p = 0.039, dependent t-test; Fig 8A) with maximum values of 50.0 Nm and 35.4 Nm, respectively. Torsional stability increased with higher maximum impaction forces (not significant, p = 0.085, $r^2$ = 0.326, Pearson correlation, Fig 8B). No correlation was found between trabecular or cortical BMD and maximum torsional moments ($p_{trabecular}$ = 0.909, $p_{cortical}$ = 0.872, Pearson correlation). The maximum rotation of the implant prior to failure between implant and bone due to torsional loading did not show significant differences between the two implant designs (Reclaim: 0.62 ± 0.47˚, Prototype: 0.68 ± 0.25˚; p = 0.812, dependent t-test).

## Contact analysis

The deviation between the final implant position achieved and the pre-planned position was small. The Prototype design tended to not fully seat, whereas the Reclaim design migrated deeper as intended, but the differences were not significant (Prototype: 1.2 ± 3.1 mm vs. Reclaim: -0.9 ± 4.6 mm; p = 0.496, dependent t-test). No difference in angular misalignment

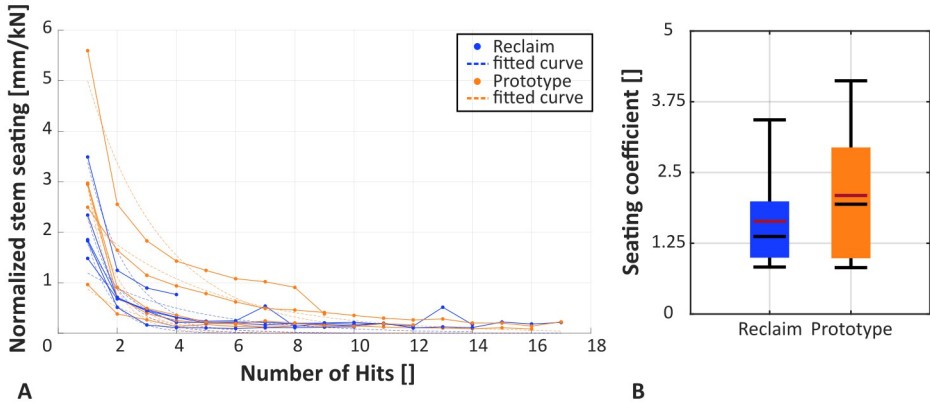

**Fig 7.** (A) Seating process for all specimens with fitted exponential curves. Some DIC measurements failed due to detached markers. (B) Stem seating coefficient for the two stem designs. Median is depicted in black and the mean is shown in red. The mean seating coefficient was higher (not significant, p = 0.305) for the Prototype design, indicating slower seating.

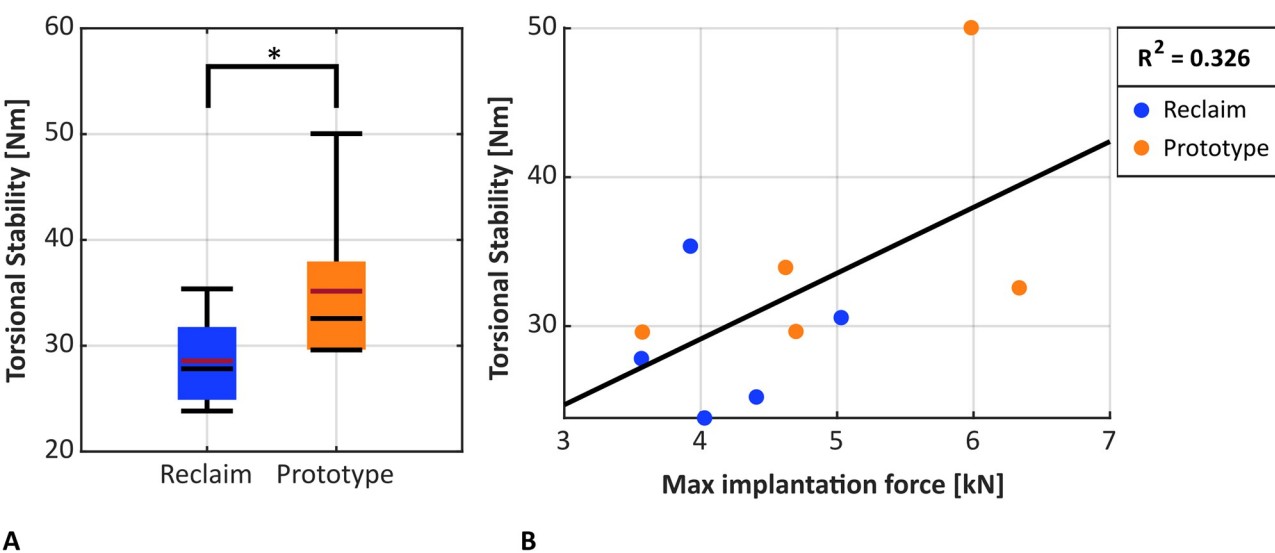

**Fig 8.** (A) The Prototype implants withstood a higher torsional moment to failure (p = 0.039). The Median is shown in black, the mean in red. (B) Torsional stability increased with maximum impaction forces (not significant, p = 0.326).

between the reamed cortical cavity axis and the implant axis was observed after implantation for either design (p = 0.393, dependent t-test).

The indentation depth of the splines was slightly higher for the Reclaim design but there was less variation in indentation depth for the Prototype design than for the Reclaim design (0.242 ± 0.041 mm vs. 0.368 ± 0.268 mm; p = 0.502, dependent t-test, Fig 9A). Trabecular BMD (p = 0.268, Pearson correlation) and cortical BMD (p = 0.210, Pearson correlation) did not have a significant effect on indentation depth. Slightly longer contact was achieved with the Prototype stems compared to the Reclaim stems (11.62 ± 3.05 mm vs. 9.45 ± 2.56 mm; p = 0.474, dependent t-test). When the 5% of specimens with the longest contact lengths for each design were evaluated as a measure of maximum contact length, a similar effect was

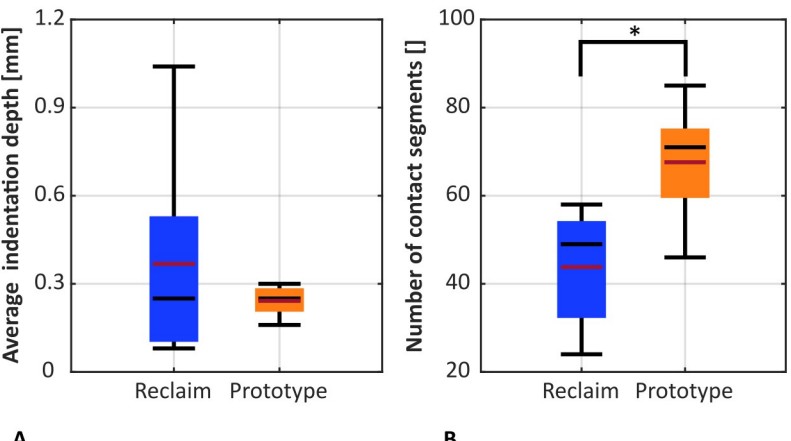

**Fig 9.** (A) Average cortical indentation depth of the longitudinal splines was not significantly different for the two designs (p = 0.502). (B) Number of matrix segments indicating cortical contact areas was higher for the Prototype design (p = 0.046). The median is shown in black, the mean in red.

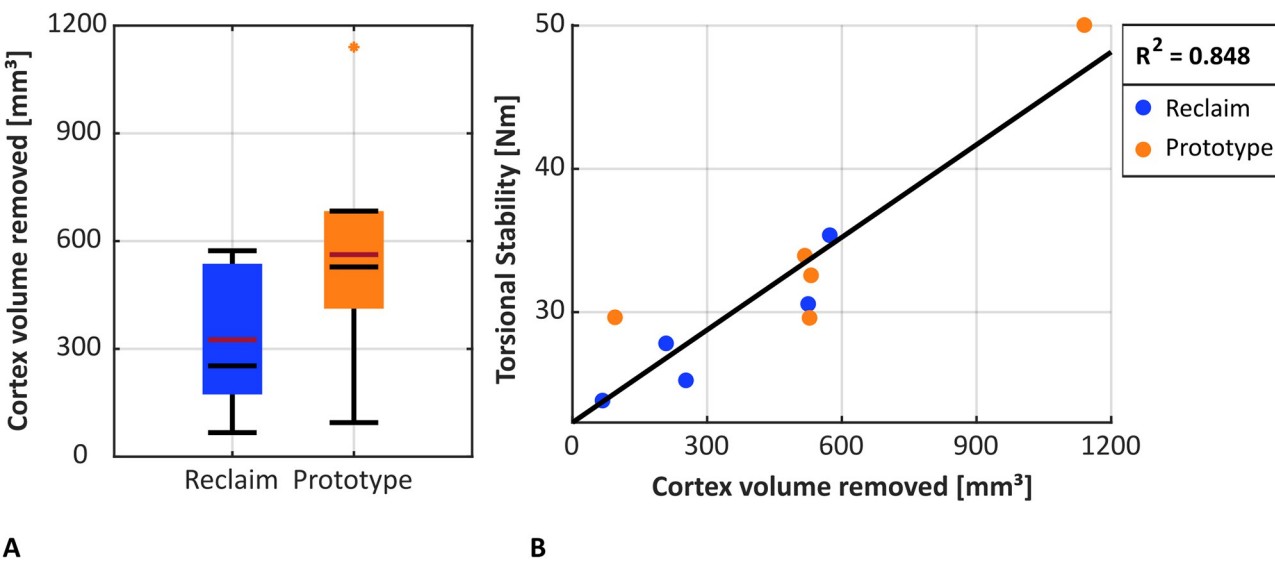

**Fig 10.** (A) Cortical bone volume removed during cavity reaming. The median is shown in black, the mean in red. (B) The torsional stability increased with bone volume removed (p < 0.001, r² = 0.848).

found (48.1 ± 10.4 mm vs. 40.9 ± 15.3 mm; p = 0.540, dependent t-test). The contact area, as measured by the number of matrix segments with contact between the stem and the cortical femoral canal, was increased by 54% for the Prototype stem compared to the Reclaim stem (p = 0.046, dependent t-test, Fig 9B).

Contact locations were quite similar for the two designs for all three analyzed regions ($p_{anterior/posterior}$ = 0.262, $p_{medial/lateral}$ = 0.522, $p_{proximal/distal}$ = 0.626, dependent t-tests).

Cortical bone volume removed during reaming was not significantly different for the Prototype design compared to the Reclaim stem (562.4 ± 333.7 cm³ vs. 325.3 ± 193.4 cm³; p = 0.158, dependent t-test, Fig 10A). Torsional stability increased significantly with the amount of cortical bone volume removed during reaming (p < 0.001, r² = 0.848, Pearson correlation, Fig 10B).

## Discussion

The purpose of this study was to evaluate the effect of a less prominent second set of splines on torsional stability in femurs with proximal bone defects. It was hypothesized that the second set of wider and less prominent splines, which engage the cortical bone after seating, would significantly increase the contact area and consequently provide more resistance to torsional loading, a property important to the long-term success of revision hip arthroplasty [8]. This hypothesis was confirmed as the cortical contact area of the prototype design with the two sets of splines was greater, as was the torsional stability. This is consistent with previous studies showing that a larger cortical contact area improves primary stability in revision THA [16, 23]. Furthermore, since the cortical contact length and depth of the splines did not differ, the idea of having a second set of splines that does not penetrate but only contacts the reamed surface of the femoral cavity was shown to have the desired effect. Both designs were able to achieve favorable radial cortical contact, rather than an inferior scratch fit or 3-point fixation [24], and were able to withstand torsional loading above that which occurs during a standard gait cycle [11].

The Prototype design required higher forces to be applied during the implantation process to achieve the desired implant position, likely due to the higher frictional force resulting from the increased bone-to-implant interface. The second set of splines in the prototype design also appears to distribute the load more effectively, potentially reducing the risk of stem subsidence and femoral fractures. In support of this hypothesis, despite higher implant forces for the prototype stems, no fractures occurred with either design, but due to the small group size, definitive conclusions have to be treated carefully and clinically confirmed. This favorable outcome may be due in part to the use of cerclage wires as a preventative measure in this study. The cerclage wires likely provided additional reinforcement and stability, minimizing the risk of bone fracture during the implantation procedure [25].

Adapting the approach of superimposing CT images with 3D laser scans to the revision stem-bone interface allowed in-depth evaluation of the contact situation within the femoral canal and detailed quantification of several parameters describing the bone-implant interface. Due to the small sample size and the general similarity of the stem designs, most of the parameters compared between the two designs showed small differences, potential trends and no significant differences. An interesting but unexpected finding of this study was the strong positive correlation of torsional stability with the amount of cortical bone removed during the reaming process. This effect may be explained by the non-homogeneous density gradient across the cortical cross-section with denser bone towards the outer surface of the femoral shaft [26–28].

The thin splines were shown to actually cut into the femoral cortex for both designs. The seating process for the prototype design is complete when the wider, shorter splines engage and cause a large increase in contact area rather "abruptly", confirming the design rationale. Indentation depth evaluation confirmed that the wider splines did not cut into the femoral cortex. Seating in the established design was slightly deeper with a slightly higher indentation depth of the sharp splines, but due to the small sample size, both of these findings were not significant. Despite this small difference in cortical indentation depth, the torsional load to failure was higher for the prototype design due to the contact of the wider splines. This may be due to the stopping of the seating process without a definitive stop when further indentation of the splines would require very high forces.

## Conclusions

Cortical contact is probably the most important factor in the immediate postoperative stability of revision stems. The addition of a second set of less protruding and wider longitudinal splines to a revision stem can improve the contact area between the implant and the bone, thus providing greater primary torsional and axial stability. Further improvements may be possible following this design rationale.

## Supporting information

**S1 Data. Raw study data containing all the data needed to recreate plots from the manuscript.**
(CSV)

## Acknowledgments

The authors thank Depuy Synthes for providing prosthetic components and surgical instruments.

## Author Contributions

**Conceptualization:** Julius M. Boettcher.

**Data curation:** Julius M. Boettcher.

**Formal analysis:** Julius M. Boettcher.

**Investigation:** Julius M. Boettcher.

**Methodology:** Julius M. Boettcher.

**Resources:** Benjamin Ondruschka.

**Software:** Julius M. Boettcher.

**Supervision:** Kay Sellenschloh, Gerd Huber, Michael M. Morlock.

**Writing – original draft:** Julius M. Boettcher.

**Writing – review & editing:** Gerd Huber, Michael M. Morlock.

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
