## [Decision Letter · Decision Letter 0]

14 Aug 2023

PONE-D-23-20774The influence of hip revision stem spline design on the torsional stability in the presence of major proximal bone defectsPLOS ONE

Dear Dr. Boettcher,

Thank you for submitting your manuscript to PLOS ONE. After careful consideration, we feel that it has merit but does not fully meet PLOS ONE’s publication criteria as it currently stands. Therefore, we invite you to submit a revised version of the manuscript that addresses the points raised during the review process.

ACADEMIC EDITOR: Please consider remarks indicated by the reviewers in the final version of the paper.

We look forward to receiving your revised manuscript.

Kind regards,

Pawel Klosowski, D.Sc.

Academic Editor

PLOS ONE

2. Please amend your authorship list in your manuscript file to include author Kay Sellenschloh.

4. We note that Figures 1 and 3 in your submission contain copyrighted images. All PLOS content is published under the Creative Commons Attribution License (CC BY 4.0), which means that the manuscript, images, and Supporting Information files will be freely available online, and any third party is permitted to access, download, copy, distribute, and use these materials in any way, even commercially, with proper attribution. For more information, see our copyright guidelines: http://journals.plos.org/plosone/s/licenses-and-copyright.

1. You may seek permission from the original copyright holder of Figures 1 and 3 to publish the content specifically under the CC BY 4.0 license.

Additional Editor Comments:

Please consider the reviewers remarks in the final version of the paper.

Reviewers' comments:

Reviewer's Responses to Questions

**Comments to the Author**

1. Is the manuscript technically sound, and do the data support the conclusions?

Reviewer #1: Yes

Reviewer #2: Yes

2. Has the statistical analysis been performed appropriately and rigorously? 

Reviewer #1: I Don't Know

Reviewer #2: Yes

3. Have the authors made all data underlying the findings in their manuscript fully available?

Reviewer #1: Yes

Reviewer #2: Yes

4. Is the manuscript presented in an intelligible fashion and written in standard English?

Reviewer #1: Yes

Reviewer #2: Yes

5. Review Comments to the Author

Reviewer #1: Many thanks to the Authors for their contribution. The Authors compared a novel revision stem with different splines to a standard off the shelf revision stem, using 5 pair of cadaveric femurs. the Authors noticed a better stability for the new stem. I must admit that the sbuject of the study is uptodate and interesting, but I have few concerns.

Line 35: p 0.046 should be considered a borderline value with no strong significance. I would rephrase considering your results not significant. The same all long the manuscript

Line 44-52: I would simplify this introduction, with less data and more focused concepts

Line 106: the engineer is a drawback of the study. Proptotype also positioned by the same engineer?

Line 205 Dorr C is a drawback.

Line 207: if there is a non-significant p value, please be more cautious and do not talk about trend or similarity

Line 238: p value?

Line 245: greater is too much, see above

Line 280: novel approach? Is it the first time this approach was adopted to measure? Coudl you provied some validation or comparisons to current standards?

Discussion and conclusions to be modified according to the results modifications

Reviewer #2: Overall this is a great and useful article, which will enhance the knowledge of revision hip arthroplasty. Please find my comments below:

Line 54: This sentence about aseptic loosening should be rephrased, because as it is, it is incorrect.

Original: „Aseptic loosening is one of the major causes for hip re-revisions [3], which particularly occurs when adequate resistance to subsidence or torsional relative motion is not achieved postoperatively [9, 10]

Recommendation „ Aseptic loosening is one of the major causes for hip re-revisions [3], which usually occurs in the second or third decade following total hip replacement due to a biological process that weakens the bone-implant interface, mainly due to the immunological effect of wear particles (particle disease). Early aseptic loosening, is an avoidable complication, which occurs, when adequate resistance to subsidence or torsional relative motion is not achieved postoperatively[9, 10].

Line 62:

The following sentence might be confusing to orthopaedic surgeons as it is somewhat vague.

„The suggested length over which the implant should engage with the bone ranges from 10 mm to 80 mm [15, 16].”

The two articles quoted are 25 years apart and they talk about 2 different uncemented stem anchoring phiosophies.Yes, the scratch fit philosophy (cylindrical stems) requires long bone to implant contact, whilst with the taper fluted design it can be as little as 3 cms.

Recommendation:

The suggested length over which the implant should engage with the bone ranges from as little 20 mm for taper flutted designs, to as long as 80 mm for cylindrical uncemented femoral stems[15, 16].

Figure 2 a and 2 b are missing. These pictures are important for the reader to understand the difference better. How many splines are there for the prototype? 16? (Reclaim has 8).

Line 102-105

It is not clear how the reaming was performed. Was it hand reaming or power reaming, or perhaps both? Was the final position determined by feel for the cortical contact and establishing a femoral cone fit? Was the reaming performed with the same reamers for both stem designs? This is an important issue.

Line 244-246

The contact area, as measured by the number of matrix segments with contact between the stem and the cortical femoral canal, was 54 % greater for the Prototype stem compared to the Reclaim stem (p = 0.046, dependent t-test, Fig 9B).

Comment: the contact pattern as seen on Figure 6 shows two distinctly differenct „contact maps”, the prototype seems (Figure 6A) to have contact at the distal point of the ETO then, a large contact „patch” distally between 120-240 degrees. The Reclaim stem shows a more even distribution of contact points (Figure 6B) although the areas are smaller. How do the investigatiors explain this? Are there any other differences between the stem designs (shape, tip of the stem, bevel?)? Am I interpreting Figure 6 correctly?

Line 275

„In support of this hypothesis, despite higher implant forces for the prototype stems, no fractures occurred with either design”

Although this sentence might be techincally correct, it should be interpreted with caution as there were only 10 femurs, and in vivo use of the implant might still provide suprises.

6. PLOS authors have the option to publish the peer review history of their article (what does this mean?). If published, this will include your full peer review and any attached files.

Reviewer #1: No

Reviewer #2: **Yes: **Dr Krisztian Sisak

---

## [Author Response · Author response to Decision Letter 0]

17 Aug 2023

Thank you very much for the effort you put into your review and the helpful suggestions for improvements. We have tried to accommodate them as good as possible.

Responses to academic editor

The requirements were checked in accordance with the guidelines.

2. Please amend your authorship list in your manuscript file to include author Kay Sellenschloh.

Thank you for bringing this to our attention! Authorship list in the manuscript has been completed with the inclusion of Mr. Sellenschloh.

Supporting information files were named accordingly

"S1 Raw study data. Raw study data containing all the data needed to recreate plots from the manuscript. "

4. We note that Figures 1 and 3 in your submission contain copyrighted images. All PLOS content is published under the Creative Commons Attribution License (CC BY 4.0), which means that the manuscript, images, and Supporting Information files will be freely available online, and any third party is permitted to access, download, copy, distribute, and use these materials in any way, even commercially, with proper attribution. For more information, see our copyright guidelines: http://journals.plos.org/plosone/s/licenses-and-copyright.

Thank you for the copyright check. However, all the images were created by us. Therefore, no copyright has been infringed and no permission is needed. We do not fully understand the copyright claim and believe that this comment may be a missunderstanding.

The reference list was reviewed and no papers were found to be retracted. Original reference [2] Grimberg A, Lützner J, Melsheimer O, Morlock M, Steinbrück A. EPRD - Jahresbericht 2021. 2021. Auflage. Berlin: EPRD Deutsche Endoprothesenregister 2021. Was excluded due to the simplification of the introduction as requested by reviewer #1.

Responses to reviewer #1

Reviewer #1: Many thanks to the Authors for their contribution. The Authors compared a novel revision stem with different splines to a standard off the shelf revision stem, using 5 pair of cadaveric femurs. the Authors noticed a better stability for the new stem. I must admit that the subject of the study is up to date and interesting, but I have few concerns.

Thank you for this assessment, we will try to overcome your concerns

Line 35: p 0.046 should be considered a borderline value with no strong significance. I would rephrase considering your results not significant. The same all long the manuscript.

We understand your concern about significance. We have adjusted the wording accordingly and removed the term "significant" as suggested by you. We would still like to point out that it is common practice in hypothesis testing to set a level for the size of the type I error and to reject the hypothesis if the p-value of a test is less than the set level (0.05 in our study), regardless of how much less the p-value is. With relatively small group sizes and the high variability of biological tissues, it is always difficult to achieve very low p-values for the test.

Line 44-52: I would simplify this introduction, with less data and more focused concepts

We followed your suggestion to simplify the introduction and focus on the threat posed by the number of revisions due to lack of bone stock and the resulting challenge for surgeons.

"The german arthroplasty registry EPRD documents 177,826 primary THR and 18,145 revision total hip replacements (THR) for 2022 [1], while the National Joint Registry (NJR) for England, Wales, Northern Ireland, the Isle of Man and Guernsey reports, 135.000 revision THRs for the time period between 2003 and 2021 [2]. These high numbers of revisions pose a challenge for surgeons due to reduced bone stock and increasing patient age with declining bone quality [3–6]. With each successive revision, the risk of subsequent revision increases [7]."

Line 106: the engineer is a drawback of the study. Proptotype also positioned by the same engineer?

Thank you for bringing this to our attention. The description of the implantation process has been amended accordingly to better explain the procedure and the reasoning behind it. The entire experiment was performed by the same engineer who had been trained in the bone preparation procedure by an experienced surgeon. Implantation was performed with a drop tower to eliminate variability caused by hammer blows that would otherwise have been introduced. We fully agree that surgeon experience is critical for primary implantation, but for the revision situation investigated in this study with a standardised bone defect and the engineer's training and experience with in vitro studies, the results would not have benefited from a surgeon using the drop tower. The rephrased text:

"Stem implantation was performed using a droptower (Fig. 3) for both stem designs by an engineer trained by an experienced surgeon to eliminate variability caused by mallet blows."

Line 205 Dorr C is a drawback.

We actively pursued the provision of a poor bone quality specimen set, such as Dorr Type B & C, to represent the difficult patient situation indicative of the revision stems used in this study. We do not fully understand why you consider this to be a drawback.

Line 207: if there is a non-significant p value, please be more cautious and do not talk about trend or similarity

Thank you! We have rephrased "similar" to "not significantly different" and "trend" to "higher but not significant".

Line 238: p value?

The wording has been changed to directly state the values and the corresponding p-value.

"The indentation depth of the splines was slightly higher for the Reclaim design but there was less variation in indentation depth for the Prototype design than for the Reclaim design (0.242 ± 0.041 mm vs. 0.368 ± 0.268 mm; p = 0.502, dependent t-test, Fig 9A). "

Line 245: greater is too much, see above

We agree that we should not emphasise the difference too much. The wording has been changed to be more descriptive when listing the results.

"The contact area, as measured by the number of matrix segments with contact between the stem and the cortical femoral canal, was increased by 54 % for the Prototype stem compared to the Reclaim stem (p = 0.046, dependent t-test, Fig 9B)."

Line 280: novel approach? Is it the first time this approach was adopted to measure? Could you provide some validation or comparisons to current standards?

Thank you for pointing that out. What was new was the amendment relating to quantification of the bone-implant interface and the creation of contact matrices. The wording was changed accordingly.

" Adapting the approach of superimposing CT images with 3D laser scans to the revision stem-bone interface allowed in-depth evaluation of the contact situation within the femoral canal and detailed quantification of several parameters describing the bone-implant interface."

Discussion and conclusions to be modified according to the results modifications

Done!

Responses to reviewer #2

Reviewer #2: Overall this is a great and useful article, which will enhance the knowledge of revision hip arthroplasty. Please find my comments below:

Thank you for this assessment!

Line 54: This sentence about aseptic loosening should be rephrased, because as it is, it is incorrect.

Original: „Aseptic loosening is one of the major causes for hip re-revisions [3], which particularly occurs when adequate resistance to subsidence or torsional relative motion is not achieved postoperatively [9, 10]

Recommendation „ Aseptic loosening is one of the major causes for hip re-revisions [3], which usually occurs in the second or third decade following total hip replacement due to a biological process that weakens the bone-implant interface, mainly due to the immunological effect of wear particles (particle disease). Early aseptic loosening, is an avoidable complication, which occurs, when adequate resistance to subsidence or torsional relative motion is not achieved postoperatively [9, 10].

Thank you! The sentence has been reworded following your recommendation:

"Aseptic loosening is one of the major causes for hip re-revisions [2] which historically occurred in the second or third decade following primary total hip arthroplasty due to a biological process that weakens the bone-implant interface, mainly due to the immunological effect of wear debris (particle disease). Early aseptic loosening is a preventable complication that occurs when adequate resistance to subsidence or relative torsional motion is not achieved postoperatively [8, 9]."

Line 62:

The following sentence might be confusing to orthopaedic surgeons as it is somewhat vague.

„The suggested length over which the implant should engage with the bone ranges from 10 mm to 80 mm [15, 16].”

The two articles quoted are 25 years apart and they talk about 2 different uncemented stem anchoring philosophies. Yes, the scratch fit philosophy (cylindrical stems) requires long bone to implant contact, whilst with the taper fluted design it can be as little as 3 cms.

Recommendation:

The suggested length over which the implant should engage with the bone ranges from as little 20 mm for taper flutted designs, to as long as 80 mm for cylindrical uncemented femoral stems [15, 16].

Thank you for the useful recommendation to improve our paper, the sentence has been reworded accordingly.

Figure 2 a and 2 b are missing. These pictures are important for the reader to understand the difference better. How many splines are there for the prototype? 16? (Reclaim has 8).

Sorry. The figure was included in the submission but labelled as supplemental, this has now been corrected. And yes: 16 for the prototype. The numbers were added in the paper

Line 102-105

It is not clear how the reaming was performed. Was it hand reaming or power reaming, or perhaps both? Was the final position determined by feel for the cortical contact and establishing a femoral cone fit? Was the reaming performed with the same reamers for both stem designs? This is an important issue.

Sorry for not describing this more accurately. Was rephrased:

"Powered reaming to the depth determined during planning was performed on both implants according to the respective surgical technique using respective straight helical reamers corresponding to each of the stem designs. The final reaming position was determined by the markings on the instruments."

Line 244-246

“The contact area, as measured by the number of matrix segments with contact between the stem and the cortical femoral canal, was 54 % greater for the Prototype stem compared to the Reclaim stem (p = 0.046, dependent t-test, Fig 9B).”

Comment: the contact pattern as seen on Figure 6 shows two distinctly differenct „contact maps”, the prototype seems (Figure 6A) to have contact at the distal point of the ETO then, a large contact „patch” distally between 120-240 degrees. The Reclaim stem shows a more even distribution of contact points (Figure 6B) although the areas are smaller. How do the investigatiors explain this? Are there any other differences between the stem designs (shape, tip of the stem, bevel?)? Am I interpreting Figure 6 correctly?

You are interpreting Figure 6 correctly. As a result of the double number of splines in the prototype design, the contact appears to be more of a "patch" pattern, with the wider splines occasionally contacting the cortex but preventing further indentation in that direction. In addition, small deviations between two femurs in a pair in terms of reaming accuracy or morphological differences can't be completely ruled out, and the different stem lengths may also contribute to differences in cortical contact.

Line 275

„In support of this hypothesis, despite higher implant forces for the prototype stems, no fractures occurred with either design”

Although this sentence might be technically correct, it should be interpreted with caution as there were only 10 femurs, and in vivo use of the implant might still provide suprises.

Well said! The wording has been adjusted to include the small group size to make the statement more cautious.

"The second set of splines in the prototype design also appears to distribute the load more effectively, potentially reducing the risk of stem subsidence and femoral fractures. In support of this hypothesis, despite higher implant forces for the prototype stems, no fractures occurred with either design, but due to the small group size, definitive conclusions have to be treated carefully and clinically confirmed."

---

## [Decision Letter · Decision Letter 1]

5 Sep 2023

The influence of hip revision stem spline design on the torsional stability in the presence of major proximal bone defects

PONE-D-23-20774R1

Dear Dr. Boettcher,

We’re pleased to inform you that your manuscript has been judged scientifically suitable for publication and will be formally accepted for publication once it meets all outstanding technical requirements.

Kind regards,

Pawel Klosowski, D.Sc.

Academic Editor

PLOS ONE

Additional Editor Comments (optional):

Reviewers' comments:

Reviewer's Responses to Questions

**Comments to the Author**

1. If the authors have adequately addressed your comments raised in a previous round of review and you feel that this manuscript is now acceptable for publication, you may indicate that here to bypass the “Comments to the Author” section, enter your conflict of interest statement in the “Confidential to Editor” section, and submit your "Accept" recommendation.

Reviewer #1: All comments have been addressed

Reviewer #2: All comments have been addressed

2. Is the manuscript technically sound, and do the data support the conclusions?

Reviewer #1: Yes

Reviewer #2: Yes

3. Has the statistical analysis been performed appropriately and rigorously? 

Reviewer #1: Yes

Reviewer #2: Yes

4. Have the authors made all data underlying the findings in their manuscript fully available?

Reviewer #1: Yes

Reviewer #2: Yes

5. Is the manuscript presented in an intelligible fashion and written in standard English?

Reviewer #1: Yes

Reviewer #2: Yes

6. Review Comments to the Author

Reviewer #1: Concerns were addressed. In particular, the emphasis on significance was reduced. The article deserves to be published.

Reviewer #2: I am still unable to find Figure 2. Otherwise all questions have been answered and all reviewer comments addressed.

7. PLOS authors have the option to publish the peer review history of their article (what does this mean?). If published, this will include your full peer review and any attached files.

Reviewer #1: **Yes: **Francesco Castagnini

Reviewer #2: No

---

## [Editor Report · Acceptance letter]

6 Sep 2023

PONE-D-23-20774R1 

The influence of hip revision stem spline design on the torsional stability in the presence of major proximal bone defects 

Dear Dr. Boettcher:

I'm pleased to inform you that your manuscript has been deemed suitable for publication in PLOS ONE. Congratulations! Your manuscript is now with our production department. 

Kind regards, 

on behalf of

Prof. Pawel Klosowski 

Academic Editor

PLOS ONE